# Commentary: Is Wearable Fitness Technology a Medically Approved Device? Yes and No

**DOI:** 10.3390/ijerph20136230

**Published:** 2023-06-27

**Authors:** Jennifer L. Scheid, Jennifer L. Reed, Sarah L. West

**Affiliations:** 1Department of Physical Therapy, Daemen University, Amherst, NY 14226, USA; 2Exercise Physiology and Cardiovascular Health Lab, Division of Cardiac Prevention and Rehabilitation, University of Ottawa Heart Institute, Ottawa, ON K1Y 4W7, Canada; 3School of Epidemiology and Public Health, Faculty of Medicine, University of Ottawa, Ottawa, ON K1Y 4W7, Canada; 4School of Human Kinetics, Faculty of Health Sciences, University of Ottawa, Ottawa, ON K1Y 4W7, Canada; 5Department of Kinesiology, Trent University, Peterborough, ON K9L 0G2, Canada; 6Department of Biology, Trent University, Peterborough, ON K9L 0G2, Canada

**Keywords:** wearable technology, fitness, heart rate, atrial fibrillation

## Abstract

Wearable technologies, i.e., activity trackers and fitness watches, are extremely popular and have been increasingly integrated into medical research and clinical practice. To assist in optimizing health, wellness, or medical care, these devices require collaboration between researchers, healthcare providers, and wearable technology companies in order to clarify their clinical capabilities and educate consumers on the utilities and limitations of the wide-ranging wearable devices. Interestingly, activity trackers and fitness watches often track both health/wellness and medical information within the same device. In this commentary, we will focus our discussions regarding wearable technology on (1) defining and explaining the technical differences between tracking health, wellness, and medical information; (2) providing examples of health and wellness compared to medical tracking; (3) describing the potential medical benefits of wearable technology and its applications in clinical populations; and (4) elucidating the potential risks of wearable technology. We conclude that while wearable devices are powerful and informative tools, further research is needed to improve its clinical applications.

## 1. Introduction

In a technology-driven era, wearable devices have swiftly found their place in research and clinical practice. A wearable technology is an electronic device that is typically worn on the wrist, chest, arm, or hip. In a global survey of fitness trends, wearable technology has been a top trend since 2016, and specifically the #1 reported fitness trend in 2016, 2017, 2019, and 2020 [1]. Recently, the American College of Sports Medicine surveyed more than 4500 fitness professionals and reported that wearable technology remained the top fitness trend of 2023 [2]. Wearable technology includes activity trackers and fitness watches that are manufactured by companies including Fitbit, Amazfit, Garmin, Samsung, Whoop, Apple, Misfit, and Polar. These devices offer several health- and wellness-related capabilities beyond traditional measures of physical activity (accelerometry) including heart rate, and location (Global Positioning System, GPS). Fitness watches have multiple features that track health/wellness and medical information, for example, counting steps, physical activity minutes, sitting time, heart rate, sleep duration, sleep consistency, body temperature, oxygen saturation (SpO2), irregular heart rhythms, and more. With the expansion of wearable technology from only measuring physical activity to the tracking of health/wellness and medical information, understanding the definitions and limitations of health/wellness vs. medical information is timely and important. In this commentary, we will focus our discussions regarding wearable technology on (1) defining and explaining the technical differences between tracking health, wellness, and medical information, (2) providing examples of health and wellness compared to medical tracking, (3) describing the potential medical benefits of wearable technology and its applications in clinical populations, and (4) elucidating the potential risks of wearable technology.

## 2. What Is the Difference between Tracking Health and Wellness vs. Medical Information?

The World Health Organization defines health as “a state of complete physical, mental, and social well-being and not merely the absence of disease” [3], while wellness is typically defined as the active pursuit of health. Medicine is defined as “the science and art dealing with the maintenance of health and the prevention, alleviation, or cure of disease” [4]. It is important to inform, and educate, wearable technology users that health/wellness data and medical information may be difficult to differentiate since these devices often track these variables within the same application. Health and wellness information should not be used to diagnose medical conditions, but rather it simply provides a real-time measurement of physical activity, sedentary time, and/or sleep parameters. Alternatively, medical devices or medical-related components within wearable technology must be approved by the U.S. Food and Drug Administration (FDA) in the United States, by Health Canada in Canada, or by the European Medicines Agency in Europe to perform a specific medical task related to a medical issue. For example, some of the most advanced fitness watches to date include features that monitor and flag medical information such as irregular heart rhythms (e.g., identifying episodes of atrial fibrillation; AFib).

Distinguishing between health/wellness and medical information is outlined on the product details of devices including those by Fitbit and Apple. For example, Apple specifies that measurements related to the blood oxygen sensor, electrical heart sensor, and optical heart sensors on the Series 7 Apple Watch include some components that are approved as a medical device, and some that are not. Specifically, according to the Apple online manual: “Blood Oxygen app measurements are not intended for medical use, including self-diagnosis or consultation with a doctor, and are only designed for general fitness and wellness purposes” [5], while the electrocardiogram (ECG) feature on the Series 7 Apple Watch is listed as a medical device and its limitations are clearly defined: “ECG app and irregular rhythm notification require the latest version of watchOS and iOS, and are not intended for use by people under 22 years old… The irregular rhythm notification is not designed for people who have been previously diagnosed with AFib” [5]. It is further indicated that the SpO2 measurements are not intended to diagnose medical issues, but rather they are meant for users to track their personal information. However, the ECG app has been approved as a medical device to flag for the presence of AFib.

Similar to Apple, SpO2 measurements provided by Fitbit products are considered health/wellness information. Fitbit lists SpO2 and resting heart rate measures as health metrics, and it is clear that data on the health metrics dashboard “are not intended to diagnose or treat any medical condition and should not be relied on for any medical purposes. It is intended to provide information that can help you manage your well-being. If you have any concerns about your health, please talk to a healthcare provider” [6]. Additionally, wrist-worn products contain sensors which are used to discriminate between AFib and normal sinus rhythm, and Fitbit has two medical features that are available on several of their devices for such distinction. The first feature is an alert if the watch detects an irregular heart rhythm when a user is motionless or sleeping (flagging signs of AFib) and the second is an ECG recording that can be completed at any time, which will identify episodes of AFib [7]. Both health/wellness and medical information collected by wearable technology should be used in combination with advice from a healthcare provider, especially when investigating a new, and/or managing an existing, medical condition.

Understanding the difference between health/wellness and medical data, as well as the limitations of the data collected by wearable technology may be difficult for everyday users without medical training. For example, as described above, Apple and Fitbit have an approved medical feature that screens for AFib. However, AFib is not the only irregular heart rhythm that exists; but, it is the only arrythmia for which the technology screens. It should be clear to users that a lack of data (i.e., not flagging for other irregular heart rhythms such as atrial flutter, ventricular tachycardia, ventricular fibrillation, premature ventricular contractions, etc.) does not indicate the absence of heart disease. Rather, in the case of Apple and Fitbit, no indication of AFib simply means AFib is not detected, and that it cannot be generalized to indicate that the individual is in normal sinus rhythm. Additionally, Apple and Fitbit devices do not continuously monitor for AFib, rather measurements are taken intermittently. There are some new devices on the market that offer continuous ECG monitoring with cloud-based storage and analytics designed specifically for patients that need continuous ambulatory cardiac monitoring [8]. Patients with AFib are not always in AFib (depends on whether they have a diagnosis of paroxysmal, persistent, long-standing persistent, or permanent); so, even those with an existing AFib diagnosis may not receive an AFib alert. In patients diagnosed with AFib, their episodes may last minutes to days. In many patients, it is a progressive disorder, with paroxysmal (episodes of AF lasting 30 s to 7 days), persistent (AF that lasts ≥7 days, but <1 year), long-standing persistent (AF lasting ≥1 year), and lastly permanent (continuous AF for which a therapeutic decision has been made not to pursue sinus rhythm restoration) subtypes [9]. There are many nuances associated with health/wellness and medical data; the average user with varying medical conditions may not to be able to discriminate between health/wellness and medical data and may struggle to understand the limitations. Thus, manufacturers need to provide clear messaging, and healthcare providers should educate their patients about the benefits and limitations of wearable technology. Nevertheless, there are important, positive health applications for wearable technology with approved integrated medical information once an understanding of the information is realized.

## 3. Wearable Technology Tracking Medical Information

Before medical devices are approved by the FDA and/or Health Canada to be used for medical purposes, appropriately designed and powered human clinical studies must be performed (the exact details of the level of evidence required depend on the type of medical device being evaluated) [10]. Wearable technology has been evaluated to detect AFib using photoplethysmography (PPG) [11,12]. PPG is the technology in many fitness watches that use optical sensors to detect blood flow at the wrist and determine heart rate. Algorithms have been developed that determine if the heart rate is irregular from the PPG reading and send a notification if the user is experiencing AFib. In a cohort of 187,912 adults (at least 18 years old) who used activity monitors and fitness watches (Honor Band 4, Huawei Watch GT, or Honor Watch, manufactured by Huawei Technologies Co., Ltd., Shenzhen, China) to monitor heart rhythm using PPG, Guo and colleagues [11] reported that 424 participants received a “suspected AFib” notification and 87% of these participants who followed up with the research team were confirmed as having AFib. While this trial did not investigate the sensitivity or specificity of the PPG technology used in detecting AFib, it demonstrated that this technology may be a feasible approach to inform users of suspected AFib [11]. Similarly, Perez and colleagues [12] monitored 419,297 apparently healthy adults with no prior diagnosis of AFib (self-reported) for approximately 117 days, during which AFib was present in 34% of participants (and in 35% of participants 65 years of age or older). Follow-up investigations using ECG patch strips indicated a positive predictive value for irregular pulse notifications of 0.84, indicating that 84% of AFib notifications were concordant with ECG monitoring (with validity > 80% considered good) [12]. Lubitz and collaborators collected similar data from more than 455,000 adults using Fitbit smartwatches or Fitbit fitness trackers and demonstrated that the software algorithm in these devices was able to successfully detect AFib [13]; these data are currently under review with the FDA [13].

AFib is often a relatively silent disease, affecting ~1% of the worldwide population, with many undiagnosed individuals experiencing either mild and/or few symptoms [14]. However, untreated AFib is associated with an increased risk of stroke, systemic embolism, and mortality [14,15]. When AFib is detected early, physicians can prescribe medications, including anticoagulants, that are effective at preventing strokes [15]. Thus, using wearable technology to detect AFib in a user who is otherwise unaware of the presence of this condition is a clear example of the medical benefits of fitness watches and activity monitors.

Advancements in the tracking of wellness information in wearable technology are occurring rapidly, with upgraded features that can track a plethora of health and wellness variables. For example, Fitbit lists the following as health features on the Fitbit Charge 5: SpO2, blood glucose tracking (in app only), resting heart rate, heart rate variability, skin temperature variation, breathing rate, and menstrual health tracking [16]. Basic models of wearable technology (i.e., basic activity monitors) measure steps, stairs, calories, and physical activity minutes, while more advanced devices have 24 h heart rate monitoring and integrated Global Positioning Systems (GPS), which give users more detailed information regarding their exercise sessions. These devices use self-monitoring, a behavior change tool, which has shown modest results when examining increases in moderate to vigorous physical activity [17]. Recently, Larson et al. conducted a systematic review and meta-analysis of 121 studies (16,743 participants) examining the effectiveness of the use of a physical activity monitor on improving overall physical activity and/or moderate to vigorous physical activity. They reported that the current level of evidence was low for wearable technology improving physical activity, and, specifically, moderate to vigorous physical activity [17]. However, a previous systematic review (25 studies) conducted by Cheatham and colleagues suggested that physical activity monitors may assist in achieving health-related goals such as short-term weight loss, i.e., as seen in studies that were less than 6 months in duration [18]. For example, a retrospective analysis examining a 6-month weight loss study that used Fitbit activity monitors demonstrated that the number of steps per day and at least 60 min of high-intensity exercise were significant predictors of weight loss at 6 months [19]. Thus, these overweight and obese individuals achieved superior improvements in weight loss by using fitness watches to self-monitor daily steps and high-intensity exercise. Additionally, the impact that wearable technology has on assisting in achieving predetermined health goals may be stronger in middle or older adult age groups [19]. While tracking steps or physical activity minutes is not considered a medical use of wearable technology, tracking health and wellness information can have indirect health implications and medical benefits by assisting individuals in self-monitoring their health goals. For example, tracking physical activity and sleep make wearable technology users more aware of their personal metrics, and then they can help users to track these metrics over time during behavior change. It is important to understand that wearable technology allows users to view personal metrics (quickly and easily) and is a tool that may assist with behavior change. However, using wearable technology may not be itself the direct cause of behavior change.

Heart rate is another popular health/wellness feature that allows an individual using wearable technology to assess the intensity of their exercise. Devices with this feature have optical sensors that detect blood flow at the wrist to determine heart rate. An individual can use the heart rate feature during a workout to monitor their exercise intensity goal or use their wearable technology to retrospectively assess how hard they worked. These data are not without limitations. For example, heart rates during exercise may be difficult to interpret for individuals with no formal exercise science education or training. While some users may prefer to monitor heart rate in beats per minute (BPM), others may choose the data presented as the “zone” or intensity achieved. Unfortunately, the specific “zone” or intensity achieved based on the percentage of maximal heart rate can vary by device, and does not necessarily align with metabolic and physiological stress [20], and may be different from the exercise intensities defined by the American College of Sports Medicine (ACSM) [21]. For example, the ACSM defines moderate exercise as 64% to 76% of maximal heart rate [21]. This is similar, but not identical, to the cardio zone defined by Fitbit (70% to 84% of maximal heart rate) [22]. Interestingly, not all Fitbit models use the percentage of maximal heart rate to define the heart rate zone; some use heart rate reserve to define the cardio zone [22], which is an entirely different outcome to consider. As well, the Fitbit devices that calculate heart rate zones by percentage of maximum heart rate use the following equation to estimate maximal heart rate: 220-age [22], which unfortunately is not the most accurate assessment or equation available to estimate heart rate [21,23]. It is important to note that this calculated maximum heart rate is an estimate, and that true maximum heart rate could be higher or lower by 10 to 15 beats per minute [21]. Additionally, these algorithms do not consider individual variations in heart rate related to other factors such as medications (which can blunt heart rate responses), medical conditions (e.g., arrhythmias), caffeine, or stress. Therefore, it can be difficult for individuals who are not familiar with exercise physiology research to correctly interpret their exercise heart rate data. Heart rate is only one example of using wearable technology to track wellness information, but even this perceived simple physiological metric may become confusing to users of this technology as it can be impacted by multiple factors and can vary by watch company and model. For example, Apple watches have their proprietary rings (Move, Exercise, and Stand), and while the concept for users to “close their rings” is simple, the exact meaning of the rings can be difficult to interpret, especially for an individual who is using the watch with no formal physical health education or training. Tracking health and wellness personal data has a role in health, fitness, and/or behavior change, but as more data are being collected, a greater understanding of these health metrics is needed to use the technology.

## 4. Potential for Medical Benefits of Wearable Technology

While SpO2 levels have traditionally been measured by commercially available devices designed for assessing oxygen saturation, new mainstream fitness watches and activity trackers can also monitor SpO2 [5,6]. Interestingly, these devices are not currently approved by the FDA for medical purposes [6], but research studies have shown that there may be medical benefits of monitoring SpO2 using wearable technology. For example, Pipek and colleagues (2022) examined SpO2 in 100 patients with chronic obstructive pulmonary disease and interstitial lung disease and they observed a strong positive correlation (r = 0.81, *p* < 0.001) between the Apple Watch SpO2 readings and commercial medical grade pulse oximeters [24]. Further, using SpO2 to calculate estimated oxygen variation (EVO) has also been used by researchers to demonstrate potential medical benefits [25]. SpO2 is typically very stable, but EVO provides detailed assessments of low or high variations in SpO2 [26]. Yamagami and colleagues [25] measured EVO using wearable devices (Fitbits) in 23 patients with COVID-19 and demonstrated that EVO measured by activity monitors can be used to detect early COVID-19 symptom exacerbation and may be useful information for healthcare professionals treating patients with COVID-19. While SpO2 readings taken by wearable technology have not been approved by the FDA or Health Canada and is therefore not considered a medical device, researchers have begun to demonstrate that measuring blood oxygenation or SpO2 variation may have clinical importance.

Monitoring resting heart rate or peripheral temperature using wearable technology may be another tool to predict the onset of illness. Natarjan and colleagues (2020) demonstrated in 2745 participants with COVID-19 (confirmed by a positive COVID-19 laboratory test) that respiratory rate, heart rate, and heart rate variability measured using Fitbit assisted in identifying the onset of symptoms associated with COVID-19 [27]. Smarr and colleagues (2020) showed that peripheral temperature measured by the Oura ring, a commercially available wearable technology (Oura Health, Oulu, Finland), in 50 otherwise healthy adults was associated with self-reported fever [28]. Future research should continue to investigate the use of health and wellness data collected by wearable technology in predicting illness. This may be particularly important from a public health perspective for communicable diseases (COVID-19, flu, common cold, etc.), since the earlier a patient is aware of an illness, the sooner they can take measures to prevent its further spread. Beyond the benefit to public health, the earlier an individual is aware of an illness, the sooner they can seek appropriate treatment and begin symptom management (such as anti-viral medication, which needs to be started shortly after disease onset). With an expected rise in global pandemics beyond COVID-19 [29], such wearable devices may prove to be important, cost-effective, and practical healthcare tools.

Ultimately, the scope of how health/wellness and physiological variables monitored by wearable technology (such SpO2, EVO, respiratory rate, heart rate, heart rate variability, and temperature) can be used optimally by users and healthcare providers to enhance patient care is not yet well-understood. Even though these features may not be FDA/Health Canada approved for medical use, users may have an expectation and/or desire that their use be integrated as part of their care, an important consideration for current healthcare providers.

## 5. Potential Risks of Wearable Technology

There are several limitations and potential risks of using wearable technology that should be considered. As previously noted, for a specific feature of a device to be considered a medical device, that feature must be approved by the appropriate federal organization (e.g., FDA and/or Health Canada), and for that approval there must be appropriately designed and powered human clinical studies. For example, the accuracy of detecting AFib has been evaluated in large studies (previously discussed) [11,12]. However, the reliability and validity of health- and wellness-related components (e.g., steps, energy expenditure, heart rate) of wearable technology are not held to the same standard as the medical device standard and they vary among studies [30]. Fuller et al. examined 158 publications in a robust systematic review of wearable technology devices (models of Apple, Fitbit, Garmin, Mio, Misfit, Polar, Samsun, Withings, and Xiaomi) that measured steps, energy expenditure, and heart rate, and concluded that the devices were accurate in their measurement of heart rate and steps; however, no brand of wearable technology device in their literature review fell within the acceptable limitations of accuracy for energy expenditure. Additionally, the reliability and validity of wearable technology are device specific, and assessments of current devices can quickly become out of date, i.e., the research has not been able to keep pace with the updates and releases of new wearable technology products [30]. To further impede this process, the companies also do not always share their algorithms for estimating steps, heart rate, and energy expenditure. Algorithms can be updated by a company at any time, and, again, studies examining specific technology can quickly become obsolete.

Wearable technology also has a plethora of social (over-reliance on technology), ethical (data privacy and secondary use of data), and ecological issues (battery life and waste) [31]. An over-reliance on wearable technology could theoretically lead to a false sense of security, overshadowing other important health factors or neglecting to seek care from appropriate healthcare providers [31]. Wearable technology manufacturers have published privacy policies that aim to protect private consumer information; however, these practices are at risk for data breaches. Since wearable devices collect and store various metrics that may include sensitive information, i.e., location (from the GPS systems employed in some devices), this may be of concern to some users.

Tracking health or wellness data may also lead to negative health consequence especially if tracking steps, calories, or physical activity goals becomes extreme [32]. Wearable technology that monitors energy expenditure can typically be integrated with a nutrition application that can monitor energy intake. Using this wearable technology to track calories (especially with the combination of monitoring caloric intake) may trigger or exacerbate preexisting eating or obsessive psychopathologies. Plateau et al. demonstrated a positive association between using wearable technology to monitor energy expenditure and eating concern, weight concern, and dietary restraint [33]. Users of wearable technology also had increased exercise compulsions and increased purging behaviors following exercise compared to non-users of wearable technology [33]. While the research on the negative impact of wearable technology is limited, these issues should be considered by healthcare providers or fitness professionals who are recommending wearable technology to their patients/clients.

## 6. Future Directions of Medical Benefits and Clinical Populations

Stakeholders, including manufactures, researchers, healthcare providers, and users, should be involved in the positive progression of wearable technology. The manufacturers of wearable technology must continue to provide clear messaging regarding how to interpret the data from their devices, and healthcare providers should educate their patients about the benefits and limitations of wearable technology. Future research should demonstrate the utility of wearable technology in the onset and management of different illnesses and chronic diseases and in the recovery of medical procedures or injuries [34]. Future research should also continue to investigate the negative outcomes of wearable technology use, especially with emerging evidence that wearable technology could be related to disordered eating [32,33] or psychosocial consequences. Healthcare providers should consider updating their patient information guides to reflect the evolving types of wearable technology available, the accuracy of the applications associated with fitness trackers, and how the technology can be optimally leveraged. Holko and colleagues (2022) investigated the barriers to using wearable technology to improve health outcomes and noted the awareness of fitness trackers and cost as the greatest barriers to participants using wearable technology to improve their health and well-being [35]. Participants desire additional knowledge regarding how wearable technology can provide key insights into their physical health, which suggests that the potential power of this technology is not being fully leveraged [35]. Chandraskaran and colleagues (2020) surveyed 4551 adults across the United States and demonstrated that most wearable technology users were Caucasian, younger (18–50 years), with some level of college education/college graduates, and included those with annual household incomes greater than US $75,000 [36]. As wearable technology becomes increasingly integrated into healthcare, researchers should ensure that this technology is not only accessible across sex, gender, ethnicity, and socioeconomic status, but also that there is greater participant diversity within studies investigating the impact of wearable technology on health outcomes.

## 7. Conclusions

Wearable technologies, including fitness watches and activity monitors, have demonstrated medical roles (e.g., detecting AFib), but also have additional advantages that are currently being studied (e.g., monitoring SpO2 or resting heart rate during illness or disease). It is also understood that wearable technology has the potential to play a role in promoting physical activity. Since physical activity is integral in preventing and managing many chronic diseases, health and wellness tracking abilities through wearable technology could assist in the clinical management of obesity, cardiovascular, and cerebrovascular diseases. However, further research is needed to improve the clinical application of these devices. Specifically, patients, healthcare providers, and researchers need to understand the role of wearable technologies in healthcare more thoroughly, including the potential risks and negative consequences of wearable technology. Device manufacturers need to ensure clarity related to clinically proven capabilities, while healthcare professionals should be educated on the utility that wearable devices may play in their specialty. These devices are powerful and popular informative tools; the potential to appropriately leverage their use to inform clinical outcomes requires continued collaboration between researchers, clinicians, and wearable technology companies.

## Data Availability

Not applicable.

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
