# Peer review of "Commentary: Is Wearable Fitness Technology a Medically Approved Device? Yes and No"

_ijerph, 2023, doi:10.3390/ijerph20136230_

Round 1

Reviewer 1 Report

I would like to thank the authors for their submission. I have only one specific comment regarding the paper which I would like the authors to address if possible. This is a minor typo in the second last sentence in the section titled "What is the difference between tracking health and wellness vs. medical information?" on page 3. Should it not read Thus, manufacturers need..... Not Thus, manufactures need.....

Overall, a decent read with some interesting content. Personally, I feel the title could be deemed a little mis-leading as you focus on one or two aspects per section. And as the use of wearables has grown significantly, this seemed to be a little light on depth. However, as it is a commentary, it does give suitable amount of information for the reader to grasp the information and overall topic area.

Reviewer 2 Report

A significant area of scientific and clinical interest is covered in this commentary. However, as the authors note in their paper, the transition from a fitness-based application to a clinical-based application does not seem to be as straightforward and easy as the authors seem to suggest.

In order to avoid this extreme delicacy, we believe that the article should be revised by providing more references, especially in sensitive areas, in order to avoid this extreme delicacy. For example, there is no reference to support the statement that "In patients diagnosed with Afib, their episodes can last minutes to days. This is usually a progressive disorder..." as there is no evidence to support it.

The conclusions have underestimated the potential risks of fitness trackers if they are used casually in clinical settings if they are not used properly.

Nevertheless, if more attention is paid to these points, it is believed that a significant contribution to science could be maintained if more attention is paid to these points. While this contribution is not particularly innovative, it does contribute to the overall rationale of the discussion.

Reviewer 3 Report

This paper addresses Wearable Fitness Technologies and their related medical purposes. Authors have provided some general topics that can founded easily in previous studies without any special discussion or comparison. Tables and categorization are needed for demonstrating clinical barriers of these devised and clear suggestion to improve them. Thus, the paper cannot be acceptable in its current form
